# Flavonoids Regulate Inflammation and Oxidative Stress in Cancer

**DOI:** 10.3390/molecules25235628

**Published:** 2020-11-30

**Authors:** Guangxing Li, Kaiyue Ding, Yanling Qiao, Liu Zhang, Luping Zheng, Taowen Pan, Lin Zhang

**Affiliations:** Institute (College) of Integrative Medicine, Dalian Medical University, Dalian 116044, China; lgx2020.11.23@gmail.com (G.L.); crystalting97@gmail.com (K.D.); qyl930303@gmail.com (Y.Q.); zliu960904@gmail.com (L.Z.); lp.zheng@siat.ac.cn (L.Z.); pantw@dmu.edu.cn (T.P.)

**Keywords:** flavonoid, anti-cancer, inflammation, oxidant

## Abstract

Cancer is the second leading cause of death globally. Millions of persons die due to cancer each year. In the last two decades, the anticancer effects of natural flavonoids have become a hot topic in many laboratories. Meanwhile, flavonoids, of which over 8000 molecules are known to date, are potential candidates for the discovery of anticancer drugs. The current review summarizes the major flavonoid classes of anticancer efficacy and discusses the potential anti-cancer mechanisms through inflammation and oxidative stress action, which were based on database and clinical studies within the past years. The results showed that flavonoids could regulate the inflammatory response and oxidative stress of tumor through some anti-inflammatory mechanisms such as NF-κB, so as to realize the anti-tumor effect.

## 1. Introduction

In recent years, the incidence of cancer has been increasing year by year. According to the data provided by World Health Organization (WHO), in 2018, there were 18,078,957 new cancer cases and 9,555,027 deaths worldwide [1]. The top three cancers with the highest new cases are lung cancer, breast cancer, and colorectal cancer, and the top three cancers with the highest mortality rates are lung cancer, colorectal cancer, and gastric cancer. The etiology of cancer is very complex, but according to the research, the occurrence of cancer has a great relationship with chronic inflammation and oxidation. Studies have shown that 20% of cancers are linked to inflammation [2]. In general, when inflammation occurs in the body, inflammation can regulate pathological and physiological signals of the body by affecting a variety of cells and factors, so as to promote the development of balance towards tissue repair and the elimination of inflammation. At the same time, inflammation is an adaptive response of the body. However, cancer is associated with chronic inflammation [3]. Chronic inflammation can promote the development of cancer, while tumor-induced inflammation can further accelerate the development of cancer [4]. Chronic inflammation can cause cancer in a number of ways. For example, chronic inflammation is often associated with mitochondrial damage [5]. The release of reactive oxygen species (ROS) after mitochondrial injury induces mitochondrial autophagy and leads to cellular functional defects. The overexpression of inflammatory signaling molecules can enhance the anti-apoptosis ability of cells and promote cell growth. NF-κB is a key regulatory protein in the process of carcinogenesis. Research results in animal models of inflammation-related tumors, such as liver cancer, colorectal cancer, and gastric cancer, have all proven the relationship between NF-κB and inflammation and tumorigenesis [6]. 

In addition, some studies have shown that patients with malignant tumors are in a low antioxidant state and have elevated oxidative stress levels [7]. High level of ROS plays an important role in the growth, proliferation, invasion, and metastasis of tumor cells [8]. There is abundant ROS in tumor tissues which are infiltrated by inflammatory cells. The use of antioxidants in animal models can effectively block the oxidative damage of DNA caused by ROS and the occurrence of malignant tumors [9]. Meanwhile, ROS can promote the activation of procarcinogens, such as aflatoxin, estrogen, and aromatic hydrocarbons, and cause DNA damage [10,11], which exacerbates the development of cancer.

Nowadays, the treatment options for cancer are surgery, chemotherapy, radiation therapy, immunotherapy, and palliative care, but chemotherapy remains the main treatment. At present, chemotherapeutic drugs are mainly divided into two categories: cytotoxic drugs and non-cytotoxic drugs. The cytotoxic drugs mainly include cisplatin, paclitaxel, and vinblastine, etc. The non-cytotoxic drugs mainly include targeted drugs and hormone drugs, etc. However, cytotoxic drugs not only kill cancer cells, but also harm normal cells. The toxic reaction has become a factor of concern for drug dosage, and also affects the quality of life of patients. For example, bone marrow suppression, gastrointestinal reactions, and hair loss are the most common side effects. In addition, although non-cytotoxic drugs are less toxic, there are still some side effects. For example, small molecule kinase inhibitors are highly specific, but they still have gastrointestinal reactions.

In recent years, treatment of cancer with flavonoids have attracted more and more attention for its low toxic or side effects. Flavonoids now refers to a series of compounds which are connected with three carbon atoms in the center and two benzene rings with phenolic hydroxyl group. Structurally, flavonoids include 2-phenylchromans and 3-phenylchromans. The 2-phenylchromans are flavonoids which contain flavanones, flavones, flavonols, flavanol, and anthocyanidins, while 3-phenylchromans are isoflavones [12]. The structural formulae of flavonoids is shown in the Figure 1 below. According to existing researches, flavonoids have lots of benefits, such as anti-cancer [13], anti-inflammatory [14], antioxidant [15], and anti-cardiovascular disease [16], etc.

We will mention the following flavonoid substances and their structural formulas:

## 2. Classify

### 2.1. Lung Cancer

Lung cancer has the highest incidence and fatality rate worldwide, and a five-year survival rate is only about 15% at all stages [17]. Lung cancer is divided into small cell lung cancer and non-small cell lung cancer, of which non-small cell lung cancer accounts for 85% [18]. Nowadays, the treatment of lung cancer is targeted therapy, immunotherapy, and chemotherapy, of which chemical therapy is indispensable, and the chemical drugs currently used include platinum drugs, anti-metabolic drugs, vincristine (VCR), and paclitaxel, etc. At present, the recommended treatment for first-line chemotherapy is a two-drug solution containing platinum [19]. However, studies have found that chemical drugs have significant side effects, e.g., platinum has neurotoxic effects [20], which greatly limits the use of lung cancer treatment. Flavonoids derived from food can largely avoid this problem. Studies have suggested that peonidin 3-glucoside (P3G), as a kind of flavonoids, can inhibit lung cancer cells in a variety of ways, such as lowering the extracellular signal-regulated kinase (ERK) pathway to inhibit H2199 cell invasion, inhibiting the mitogen-activated protein kinase (MAPK) pathway and regulating extracellular matrix (ECM) degradation protease to inhibit the invasion and activity of lung cancer cells. Moreover, experiments have confirmed that P3G has no toxic effect [21]. Eicosanoid which produced by arachidonic acid (AA) with lysyl oxidase (LOX) metabolism is related to cancer. Baicalein can induce H460 apoptosis by inhibiting 12-LOX [22]. Kaempferol can inhibit Matrix metalloproteinase (MMP)-2 activity of human A549 lung cancer cells [23]. Icaritin can be regulated by microRNA-10a the phosphatase and tensin homolog deleted on chromosome ten (PTEN/Akt) and extracellular signal-regulated kinase (ERK) pathway to inhibit human A549 lung cancer cells [24]. Quercetin can play an anti-cancer role in a variety of ways. Youn and others believe that quercetin induces the apoptosis of SH460 cell by inhibiting NF-κB signaling pathways [25]. There have also been many experiments using Western blotting that found quercetin can reduce the expression of the MMP-2. Some studies reported that luteolin is capable of activating ERK MAPK pathway in different cells [26,27,28]. Therefore, Zhuang Hong et al. speculated that luteolin could slightly increase the activity of the Ras/Raf/MAPK/ERK pathways in double-mutant NSCLC cells [29]. Xueting Cai et al. found that luteolin completely suppressed the necrosis factor alpha (TNFa) induced the activation of nuclear factor kB (NF-κB) p65 by Western blotting [30]. Chrysin is used to cause an increase in reactive oxygen species (ROS) in A549 cells [31], and ROS can activate AMP-activated protein kinase (AMPK) [32,33].

Experiments by Hsu-Feng Lu et al. have shown that apigenin can increase the production of ROS and Ca2þ in H460 cells and alter the Bcl-2 associated X protein, and B cell lymphoma 2 (Bax/Bcl-2) [34]. Naringenin inhibits proliferation and induces apoptosis of A549 cells by inducing death receptor 5 (DR5) expression in human lung cancer cells and Naringin also be able to inhibit Akt activity and down-regulates MMP-2 and MMP-9 to inhibit migration of A549 cells [35,36]. Inhibiting synergistic activity of MAPKs/activating protein-1 (AP-1)and the inhibitor of NF-κB kinases(IKKs)/inhibitor kappa B(IκB)/NF-κB signaling pathways, naringin attenuates EGF-induced mucin 5AC(MUC5AC)secretion in A549 cells [37]. Some studies have also shown that (-)-epigallocatechin-3-gallate(EGCG)in green tea is the most biologically active and inhibits the expression of NF-κB to inhibit lung cancer [38]. C3G in anthocyanidin can inhibit AP-1, MAPK, NF-κB, and Cyclooxygenase-2 (COX-2) while inhibiting the metastasis and growth tumors [39].

### 2.2. Breast Cancer

Breast cancer is the most common malignant tumor in women, and the global cancer morbidity and mortality statistics released in 2018 show that breast cancer has the second highest incidence and fifth mortality rate in the world, accounting for nearly one-fourth of the incidence of malignant tumors in women, and the highest mortality rate [1]. Many flavonoids have therapeutic effects on breast cancer. Ampelopsin can inhibit tumor cell activity and promote apoptosis through reactive ROS and endoplasmic reticulum stress, but has no toxicity to normal mammary epithelial cells (McF-10a) [40]. Wogonin can activate extracellular regulatory protein kinase (ERK) and inhibit pi3K-Akt-Survivin pathway to cause McF-7 cell apoptosis [41]. By inhibiting the expression of COX-2 and cytochrome P450 (CYP 4A), isoliquiritigenin can reduce the secretion of Prostaglandin E2, reduce the phosphorylation level of intracellular the phosphatidylinositol 3-kinase (PI3K), phosphoinositide-dependent protein kinase 1 (PDK) and Akt, and thus inhibit the migration, invasion and apoptosis resistance of MDA-MB-231 and BT-549 cells [42]. Silibinin induces McF-7 cell death in two ways: on the one hand, it down-regulates Estrogen receptors alpha (ERα), thus increasing pro-apoptotic autophagy downstream and leading to cell death. On the other hand, pro-survival ROS/RNS are upregulated, and ROS/reactive nitrogen species (RNS) and autophagy production form a negative feedback loop whose balance is regulated by Erα [43]. Hesperetin induced apoptosis of McF-7 cells through ROS accumulation and activation of the apoptosis signal-regulating kinase 1 (ASK1)/c-Jun-N-terminal kinase (JNK) pathway [44]. Some studies have found that the daidzein can be combined with R-and S-equol to reduce MMP-2 to suppress the invasion of MDA-MB-231 human breast cancer cells [45]. In vitro cell studies have shown that genistein has anti-tumor effects through multiple pathways, such as NF-κB, JNK, and ERK signaling pathways, which may be involved in the process of genistein inducing the apoptotic signals in MDA-MB-231 cells. EGCG inhibits the metastasis of breast cancer cells by restoring the balance between MMP and the substring matrix-metalloproteinase inhibitor (TIMP) [46]. Shannaiphenol induces apoptosis by inhibiting the expression and function of the estrogen receptor ERα, lowering the expression of polo-like kinase 1 (PLK-1), and exciting continuously ERK signaling paths to inhibit the proliferation of MCF-7 cells in human breast cancer [47,48,49]. Nar is a potential PI3K inhibitor and mitogen-activated protein kinase(MEK)inhibitor, which inhibits the activity of phosphatidylinositide 3-kinase (PI3K) which is a key regulator of GLUT4 translocation induced by insulin and suppress the phosphorylation of impaired downstream signaling molecule Akt and the proliferation of McF-7 breast cancer cells by inhibiting the uptake of glucose [50].

In addition, the synthetic LW-214 down-regulates the trX-1 (Thioredoxin-1) protein then increases intracellular ROS and activates apoptosis signal-regulated kinase 1 (ASK1), leading to the activation of C-Jun amino-terminal kinase (JNK) and induce cell apoptosis through the mitochondrial pathway [51]. In studies, oncamex, a second-generation flavonoid derivative of myricetin, has a stronger anti-tumor activity than myricetin, which causes apoptosis by regulating changes in ROS in mitochondria in a variety of breast cancer cells, including MCF-7 and MDA-MB-231, etc. [52]. Synthetic LGF-500 also inhibits the invasiveness of MDA-MB-231 cells by inhibiting the PI3K/Akt/ NF-κB signaling pathway [53]. Moreover, some studies show that the IC50 value of quercetin against MCF-7 cells was 0.87 mg/mL [54].

### 2.3. Gastric Cancer

Gastric cancer is a malignant tumor that occurs in the epithelial of the stomach mucosa and the cause is complex. However, helicobacter pylori infection, environmental factor and genetic factor have become recognized as the cause. In the treatment methods, 5-Fluorouracil (5-Fu), capecitabine, and tecchio were generally selected as the chemical drugs, and the combined drug was generally more effective than the single drug. However, there is no ideal compatibility of medicines. Flavonoids play an important role in the treatment of gastric cancer.

In gastric cancer cell lines, hesperetin inhibits cell proliferation and induces apoptosis by promoting intracellular ROS accumulation [55]. The treatment of human gastric cancer cell (BGC-823) with genistein showed significant decrease in COX-2 protein levels and nuclear transcription factor NF-κB activity. Further studies demonstrated that genistein inhibited tumor angiogenesis by inhibiting NF-κB activity and COX-2 protein expression [56]. Kham et al. found that genistein significantly inhibited the NF-κB pathway in mouse skin inflammation, showing strong anti-inflammatory and anti-tumor activity [57]. Naringenin can significantly inhibit the growth of human gastric cancer cells (SGC-7901), reduce the expression of MMP-2 and MMP-9 in gastric cancer cells, and significantly suppress the proliferation, adhesion, invasion, and migration of SGC-7901 [58].

### 2.4. Colorectal Cancer

In recent years, the incidence of CRC has been on the rise in most countries in the world, especially in China where the number of new cases has been increasing at a rate of 4% year. It is generally believed that its pathogenesis is related to environmental and genetic factors. Studies have found that EGCG in tea can induce hT-29 cell apoptosis via Akt, ERK1/2 and other p38MAPK pathways [59]. The NF-κB pathway plays an extremely important role in the progression from inflammation to cancer in the colon. For example, NF-κB-mediated Notch and JNK signaling pathways are also involved in colorectal cancer progression [60]. The downstream NF-κB signaling pathway is activated when LPS binds specifically to TOLl-like receptor 4 during inflammation. NF-κB is activated when it is transferred from the cytoplasm to the nucleus, and further activates other genes that cause cell proliferation and inhibit apoptosis [61]. Quercetin has been shown to inhibit inflammatory responses and induce the apoptosis of colon cancer Caco-2 and SW-620 cell lines by inhibiting NF-κB signaling, down-regulating B cell lymphoma-2 (Bcl-2) and up-regulating Bax [62]. Some also believe that quercetin can induce apoptosis by phosphorylation of AMPK and p53 [63]. In colorectal cancer, baicalein could inhibit cell migration and invasion by reducing the expression of MMP-2 and MMP-9 via suppression of the Akt signaling pathway [64]. According to in vitro studies, the anticancer effect of genistein on colorectal cancer may be related to inhibition of Wnt and NF-κB signaling pathways [65]. Naringenin regulates the expression of cell cycle genes by down-regulating cyclin dependent kinase4 (Cdk4), Cdk6, Cdk7, Bcl-2, X-IAP, and C-IAP-2, and up-regulating the expression of p18, P19, P21, Caspas-3, 7, 8 and 9, Bak, AIF and Bax in cells, so that the cell cycle can stay in the S phase or G2/M phase, and induce the apoptosis of colorectal cancer cells (SW1116, SW837). Meanwhile, the expression levels of cell survival factors PI3K, pAkt, pIκBα, and NF-κB P65 were also decreased [66].

### 2.5. Liver Cancer

Liver cancer is the most common malignancy of the digestive system with high death rate. Some natural foods and their active ingredients can affect the occurrence and development of liver cancer by inhibiting the growth and metastasis of tumor cells. The study of Li et al. has found that Oroxin B(OB) can negatively regulate PTEN gene by down-regulating the expression of miR-221 and inactivate the PI3K/Akt signaling pathway [67], while it has been reported that PI3K/Akt pathway plays an important role in regulating inflammatory responses [68]. Baicalein was found that it is able to inhibit the proliferation of Bel-7402 cells by inducing cell cycle arrest at the S and G2/M phase via up-regulating the expression of p21 and p27 and suppressing the PI3K/Akt pathway [69]. Further, there are many pathways that baicalein can fight against hepatocellular carcinoma, such as MAPK and NF-κB pathway, ERK pathway, Akt/mammalian/mechanistic target of rapamycin (mTOR) pathway, etc. [70]. Moreover, isobavachalcone was found to exert anti-proliferative and pro-apoptotic effects on human liver cancer cells by targeting the ERKs/RSK2 signaling pathway [71]. In the study of Liu et al., after fisetin administration and bromocriptine treatment, the expression of TGF-β1 was down-regulated in liver cancer cells [72]. Meanwhile TGF-β1 is well known in tumor growth, which could be regulated for ERK signaling pathway [73] and the ERK signaling pathway activates TGF-β1 expression and leads to tumor metastasis [74]. In addition, it has been reported that, when the IC50 of quercetin is 4.88 mg/mL, the survival rate of HepG2 cell lines can be best reduced [54]. The anticancer effect of xanthohumol induces growth inhibition and apoptosis of human liver cancer through the NF-κB/p53-apoptosis signaling pathway [75]. Galangin and ellagic acid reduced the TPA-induced enzyme activity of matrix metalloproteinase-2(MMP-2) and matrix metalloproteinase-9 (MMP-9) in HepG2 cells [76,77]. It can be seen that flavonoids play an important role in inhibiting the metastasis and development of liver cancer through oxidative stress and anti-inflammatory pathways.

### 2.6. Cervical Cancer

Cervical cancer is the second most common type of cancer and the sixth leading cause of cancer-related mortality in females worldwide. Apigenin is a natural flavonoid, and Josip et al. found that it can inhibit migration and invasion in cervical cancer cells through the elevation of ROS and LPO, and mitochondrial membrane potential’s decrease [78]. Quercetin could inhibit anti-apoptotic proteins according to docking studies. Further, quercetin blocks PI3K, MAPK, and WNT pathways [79]. Genistein inhibited the growth of HeLa cells in a dose-dependent and time-dependent manner by modulating the expression of MMP-9 and TIMP-1 [80]. Xanthohumol (XN) and resveratrol uses different mechanisms to induce cell death in cell lines derived from cervical cancer, including NF-κB and P65 [81,82].

### 2.7. Prostatic Cancer

Flavonoids are natural antioxidants found in various foods. Some experimental evidence indicates that flavonoids could prevent prostate cancer. It has many anti-inflammatory or antioxidant mechanisms to promote anti-cancer effects. In the Netherlands Cohort Study, dietary flavonoid intake and black tea consumption were associated with a decreased risk of advanced stage prostate cancer [83]. Venè R et al. transgenic adenocarcinoma of the mouse prostate (TRAMP) transgenic mice was used as an in vivo model of prostate adenocarcinoma. They observed that the treatment of prostate cancer cells with low micromolar doses of xanthohumol inhibits proliferation and modulates focal adhesion kinase (FAK) and Akt phosphorylation, leading to reduced cell migration and invasion. Oxidative stress by increased production of reactive oxygen species (ROS) was associated with these effects [84]. Sanjeev et al. show that apigenin suppressed prostate tumorigenesis in transgenic adenocarcinoma of the mouse prostate (TRAMP) mice through the PI3K/Akt/FoxO-signaling pathway [85]. Quercetin is considered to be a strong antioxidant that scavenges free radicals and bind transition metal ions. With quercetin treatment, the level of lipid peroxides and H_2_O_2_ were decreased due to its scavenging properties of free radicals, and thereby inhibits prostate cancer initiation [86]. Moreover, quercetin can also increase the activation of PI3K/Akt or RAF/MEK/ERK system and AR mediated growth factor IGF-1/IGF-1R axis to inhibit the development of prostate cancer [87]. The flavonoid apigenin reduces prostate cancer CD44+stem cell survival and migration through PI3K/Akt/NF-κB signaling [88]. Naringenin-induced apoptotic cell death in prostate cancer cells is mediated via the PI3K/Akt and MAPK signaling pathways [89].

### 2.8. Other Cancer

The potential antiproliferative effects of apigenin have been recently evaluated in chemoresistant ovarian cancer cells. Apigenin decreased the viability of both parental and chemoresistant SKOV3 cells through the downregulation of TAM receptor tyrosine kinases expression. It also downregulated their downstream targets Akt and Bcl-xL [90]. Hesperetin suppressed the expression of phosphorylated PI3K/Akt, cyclin D1, MMP-2, and MMP-9 and increased phosphorylated PTEN and p21 in esophageal cancer cells [91]. Phellamurin (Phe) repressed the PI3K/Akt/mTOR pathway in osteosarcoma cells [92].

## 3. Discussion

Table 1 summarized some evidence about the use of flavonoids in cancer. It can be seen from the above that different flavonoids can regulate the inflammatory response and oxidative stress of tumors and play an anticancer role. As is shown in Figure 2 demonstrated the role of major signaling pathways The NF-κB pathway is the most common pathway for lung cancer, liver cancer and colorectal cancer. Indeed, it has been demonstrated that flavonoids suppress the expression of pro-inflammatory mediators (NF-κB cascade), have vasodilator activity, improve vascular endothelial function, protect cells against insulin resistance, regulate proliferation, and suppress neuroinflammation by reducing cytokine release [93,94,95,96,97,98,99,100]. NF-κB/Rel binds to and is inhibited by the IB protein in classical (canonical) signaling pathways. It is generally believed that, under conditions of trauma, low oxygen, and physical and chemical stimulation, the NF-κB by a large amount of activation and nuclear transport, induced by a variety of cell factors, chemical factors, adhesion molecules, enzymes, and the transcription of antimicrobial peptides, rapidly activates the immune response, which has played central role in inflammation by inhibiting the phosphorylation and degradation of κB predominate I to stop the NF-κB nuclear transfer by inhibiting the p65 subunit to the cell nuclear transfer to prevent the NF-κB and the combination of DNA, etc. The activation of NF-κB regulates the synthesis of proteins, some of which influence both inflammatory response and tumor formation. The anti-tumor effects of flavonoids are related to anti-free radicals or anti-lipid peroxidation, while most prostate cancer, lung cancer and breast cancer are ROS pathways, which are generally accompanied by cell necrotizing apoptosis. As for the relationship between ROS and tumor, some studies believe that ROS can play a role in mediating tumor cell apoptosis [101]. Breast cancer, prostate cancer, liver cancer, and esophageal cancer act on matrix metalloproteinases in another way to inhibit angiogenesis and regulate vascular endothelial growth factor. Angiogenesis is the biological process by which existing blood vessels form new ones. This is a critical process that promotes development, skeletal muscle hypertrophy, menstruation, pregnancy, and wound healing, as well as pathologies such as neovascularization (e.g., retinopathy), rheumatoid arthritis, psoriasis, AIDS/Kaposi sarcoma, and cancer (tumorigenesis) [102,103]. Tumors need a rich blood supply to grow and survive. Newborn blood vessels can make tumor cells along the basal membrane defect space and the substrate to grow around. Further, MMPs expression changes, especially with the higher expression of MMP-2 and MMP-9, and basement membrane degradation of collagen type IV ability enhancement. This is the key step in the cancer cell invasion and metastasis, as a lot of flavonoids of MMP-2 and (or) MMP-9 have an inhibitory effect. Flavonoids regulate the expression of vascular growth factor and block its signal transduction. In addition to these mechanisms, flavonoids could regulate cell apoptosis through the activation of mitogen activated protein kinase (MAPK) signal transduction pathway and protein kinase C(PKC), which also could inhibit the expression of COX-2 and reduce the induced PGE2 [104].

## 4.Conclusions

It can be seen that flavonoids can regulate the inflammatory response and oxidative stress of tumors to achieve anticancer effect. However, there are few clinical cases of using flavonoids to treat cancer. Firstly, the oral absorption of flavonoids is slow, the bioavailability is low, and the existing preparation technology of ginkgo flavonoid pellets is in the research and development stage. Secondly, the flavonoid extract is unstable, and it needs to be configured with other complexes. For example, myricetin exhibited substantial limitations, such as poor water-solubility and low stability in the body when it was administrated orally. M10 was produced by adding a hydrophilic glycosylation group and then forming a sodium salt derivative, which exhibited excellent water-solubility (>100 mg/mL) and better stability in Wistar rat plasma and liver microsomes [105]. Flavonoid drugs will be used in the clinic through technological innovation, which could alleviate the toxic and side effects and drug resistance of Western medicine. 

## Figures and Tables

**Figure 1 molecules-25-05628-f001:**
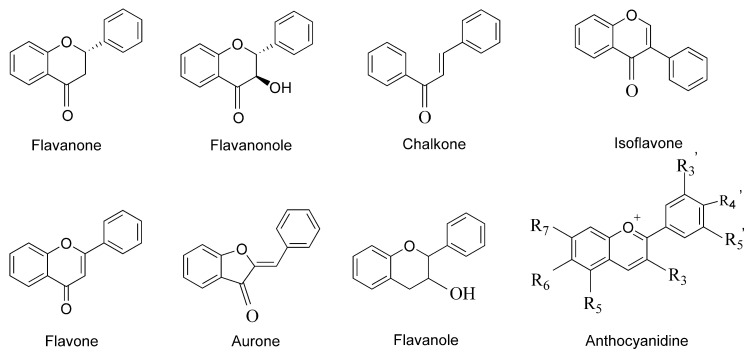
Structural formulae of flavonoids.

**Figure 2 molecules-25-05628-f002:**
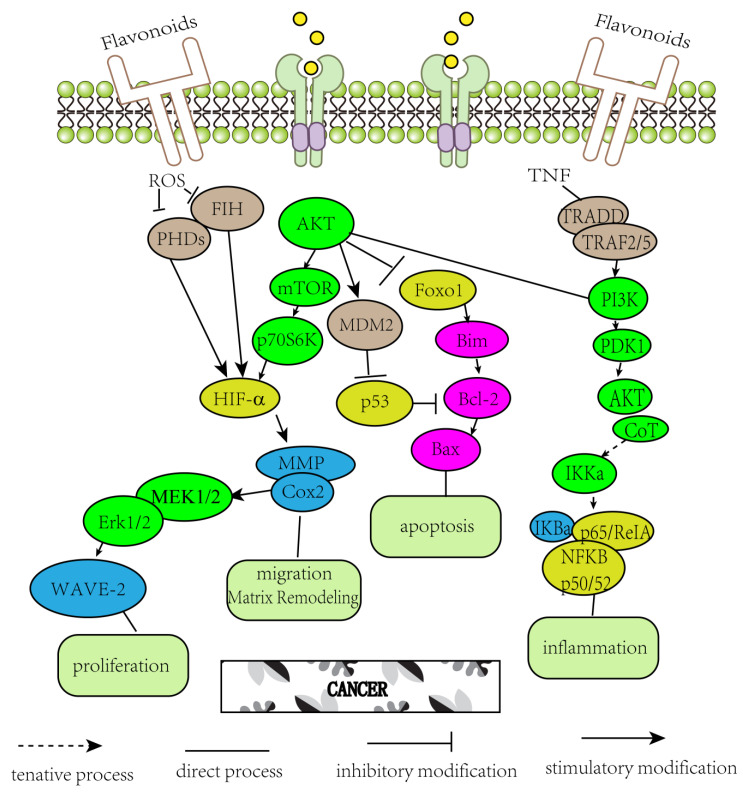
Demonstrated the role of major signaling pathways.

**Table 1 molecules-25-05628-t001:** The anti-cancer effects of flavonoids.

Flavonoid	Class	Model	Pathways	Main Effect	References
**Lung Cancer**					
Peonidin 3-Glucoside	Anthocyanidine	cell	LOX	inducing apoptosis	[21]
Kaempferol	Flavanonole	cell	MMP-2	inducing apoptosis	[23]
Epimedium flavin		cell	PTEN/Akt	inducing apoptosis	[24]
Quercetin		cell	NF-κB/MMP	inducing apoptosis	[25]
Luteolin	Flavone	cell	ERK/MAPK	MAPK pathways activation	[26,27,28,29]
Luteolin		cell	NF-kB (p65)	inducing TNF-mediated apoptotic cell death	[30]
Apigenin		cell	ROS	inducing apoptosis	[34]
Naringenin	Flavanone	cell	Akt/MMP	inhibiting tumor growth and metastasis	[35,36]
Epigallocatechin-3-gallate	Flavanole	cell	AP-1, MAPK, NF-κB, and COX-2	inhibiting tumor growth and metastasis	[38]
Baicalein	Flavone	cell	12-LOX	inducing apoptosis	[22]
Chrysin		cell	ROS	inducing oxidative stress	[31]
**Breast Cancer**					
Ampelopsin	Flavanonol	cell	ROS	promoting ER stress	[40]
Wogonin		cell	ERK	inducing apoptosis	[41]
Isoliquiritigenin	Chalkone	cell	COX-2/PI3K, PDK, Akt	inhibiting tumor growth and metastasis	[42]
Naringenin	Flavanone	cell	ERK	inhibiting tumor growth	[50]
Hesperetin		cell	ROS	inducing apoptosis	[44]
Daidzein	Isoflavone	cell	MMP	inhibiting tumor growth and metastasis	[45]
EGCG	Flavanole	cell	MMP	inhibiting tumor growth and metastasis	[46]
Kaempferol	Flavanonole	cell	PLK-1/ERK	inhibiting tumor growth and inducing apoptosis	[47,48,49]
Synthesized flavonoid LW-214	Flavone	cell	ROS	Induce cell apoptosis through mitochondrial pathway	[51]
Synthesized flavonoid Oncamex	Unknown	cell	ROS	Induce cell apoptosis through mitochondrial pathway	[52]
Synthesized flavonoid LGF-500	Flavone	cell	ROS/RNS	inducing apoptosis	[53]
**Gastric Cancer**					
Naringenin	Flavanone	cell	MMP	inhibiting chemical-induced cell invasion, metastasis	[58]
Hesperetin		cell	ROS	inhibiting cell proliferation and inducing apoptosis	[55]
Genistein	Isoflavone	cell	NF-κB/COX-2	inhibiting angiogenesis and metastasis	[56]
Genistein		mouse	NF-κB	suppressing mortality, tumor number, tumor burden and chemical-induced inflammatory responses	[57]
**Colorectal Cancer**					
EGCG	Flavanole	cell	ERK1/2/p38MAPK	inducing apoptosis	[59]
Quercetin	Flavanonole	cell	NF-κB	inducing apoptosis	[62]
Quercetin		cell	AMPK/p53	inducing apoptosis	[63]
Baicalein	Flavone	cell	MMP-2/MMP-9/Akt	inhibiting cell migration and invasion	[64]
Genistein	Isoflavone	cell	NF-κB	inducing apoptosis	[65]
Naringenin	Flavanone	cell	NF-κB/p65	inducing apoptosis and cell cycle arrest	[66]
**Liver Cancer**					
Oroxin B	Flavone	cell	PTEN/PI3K/Akt Pathway	fighting against liver cancer	[67]
Baicalein		cell	PI3K/Akt pathway	inhibiting the proliferation of Bel-7402 cells	[69]
Baicalein		cell	MAPK and NF-kB pathway, ERK pathway, Akt/mTOR pathway	fighting against hepatocellular carcinoma	[70]
Isobavachalcone	Flavanonole	cell	ERKs/RSK2 signaling pathway	anti-proliferative and pro-apoptotic effects on human liver cancer cells	[71]
Fisetin		cell	TGF-β1/ERK signaling pathway	down-regulated in liver cancer cells	[72]
Xanthohumol	Chalkone	cell	NF-κB/p53	inducing apoptosis, modulating the NF-κB/p53 and the Notch1 signaling pathways	[75]
Quercetin	Flavanonole	cell	ERK	inducing apoptosis	[76]
Quercetin		cell		suppressing chemical-induced carcinogenesis	[77]
**Cervical Cancer**					
Apigenin	Flavone	cell	ROS and LPO	oxidative stress	[78]
Quercetin	Flavanonole	cell	MAPK	decreasing cell proliferation, invasion, angiogenesis	[79]
Genistein	Isoflavone	cell	MMP-9	inducing apoptosis, cell cycle arrest, suppressing cell migration	[80]
Xanthohumol	Chalkone	cell	NF-κB	decreasing expression of CXCR4, inhibiting cell invasion induced by CXCL12	[81,82]
**Prostate Cancer**					
Xanthohumol	Chalkone	mouse	FAK/Akt/ROS	suppressing tumor growth and progression	[84]
Apigenin	Flavone	mouse	PI3K/Akt	suppressing tumor growth, angiogenesis, metastasis	[85]
Quercetin	Flavanonole	mouse	peroxides and H_2_O_2_ got decreased	inhibiting carcinogenesis induced by hormone and carcinogen	[86]
Quercetin		mouse	PI3K/Akt or RAF/MEK/ERK	inhibiting carcinogenesis induced by hormone and carcinogen	[87]
Apigenin	Flavone	cell	NF-κB/Akt	inducing apoptosis, inhibiting cell invasion, motility	[88]
Naringenin	Flavanone	cell	ERK	invasion and migration	[89]
**Ovarian Cancer**					
Apigenin	Flavone	cell	MMP	Carcinogenesis	[90]
**Esophageal Cancer**					
Hesperetin	Flavanone	cell/mouse	MMP	inhibiting chemical-induced cell invasion, metastasis,	[91]
**Osteosarcoma**					
Phellamurin	Flavanonol	cell	MMP	inhibiting tumor growth and metastasis	[92]

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
