# Peer review of "Flavonoids Regulate Inflammation and Oxidative Stress in Cancer"

_molecules, 2020, doi:10.3390/molecules25235628_

Round 1

Reviewer 1 Report

The manuscript will be of interest to the readers of this journal, after major revision to address the comments below:-

It is not clear from the abstract what the key findings are, and this needs to be included. The last sentence in the abstract is very generic/unhelpful.

Structures of the key flavonoid classes are required, either throughout the manuscript or in a suitable table in the introduction. 

More critical appraisal is needed throughout, eg to demonstrate challenges and opportunities of flavonoids - are they likely to make good drugs? How might problems be overcome?

Specific data must be provided throughout to back up the claims that are made, at the moment there is a lot of commentary but the reader is unable to validate the claims as no specific data is provided to demonstrate the efficacy/selectivity etc of the flavonoid compounds that are discussed.

Table 1 contains some interesting information but the section around it is not well developed.

There are no figures whatsoever and this is a significant weakness of the paper.

What are the conclusions? And future prospects? These sections need to be included, and they need to be robust

Author Response

亲爱的先生或女士,

Thank you so much for carefully reviewing and reconsidering our manuscript. And we would like to express our sincere thanks to the reviewers for the constructive and positive comments.

Answers to reviewers:

Reviewer #1:

Point 1: It is not clear from the abstract what the key findings are, and this needs to be included. The last sentence in the abstract is very generic/unhelpful.

Response 1:Thank you for your comments. We have already modified it in the corresponding position on page 1 line 10-18, especially the last sentence.

Cancer is the second leading cause of death globally. Millions of persons died due to cancer each year. In the last two decades, the anticancer effects of natural flavonoids have become a hot topic in many laboratories. Meanwhile, flavonoids which have been over 8000 known flavonoid molecules until now, are potential candidates for the discovery of anticancer drugs. The current review summarized the major flavonoid classes of anticancer efficacy and discussed the potential anti-cancer mechanisms through inflammation and oxidative stress action, which were based on database and clinical studies within the past years. The results showed that flavonoids could regulate the inflammatory response and oxidative stress of tumor through some anti-inflammatory mechanisms such as NF-κB, so as to realize the anti-tumor effect.

Point 2:Structures of the key flavonoid classes are required, either throughout the manuscript or in a suitable table in the introduction.

Response 2:

Thank you for your reminder. We have already added the structures on the page 2-3 line 85-94 as followed.

Point 3:More critical appraisal is needed throughout, eg to demonstrate challenges and opportunities of flavonoids - are they likely to make good drugs? How might problems be overcome?

Response 3:

 Thank you very much for pointing out this issue. Flavonoids have been innovated in the development and experiment of clinical drugs. And we added to discussion part on page 8 line 562-573 as followed:

4.Discussion

It can be seen that flavonoids can regulate the inflammatory response and oxidative stress of tumors to achieve anticancer effect. However, there are few clinical cases of using flavonoids to treat cancer. Firstly, the oral absorption of flavonoids is slow, the bioavailability is low, and the existing preparation technology of ginkgo flavonoid pellets is in the research and development stage. Secondly, the flavonoid extract is unstable, and it needs to be configured with other complexes. For example, myricetin exhibited substantial limitations, such as poor water-solubility, and low stability in body when it was administrated by oral.M10 was produced by adding a hydrophilic glycosylation group and then forming a sodium salt derivative, which exhibited excellent water-solubility (>100 mg/mL), and better stability in Wistar rat plasma and liver microsomes.[105] Flavonoid drugs will be used in clinic through technological innovation, which could alleviate the toxic and side effects and drug resistance of western medicine.

[105]Zhu,S. Yang,C. Zhang,L. Wang,S. Ma,M. Zhao,J. Song,Z. Wang,F. Qu,X. Li,F. Li,W. Development of M10, myricetin-3-O-β-d-lactose sodium salt, a derivative of myricetin as a potent agent of anti-chronic colonic inflammation. Eur J Med Chem 2019,174,9-15.

Point 4:Specific data must be provided throughout to back up the claims that are made, at the moment there is a lot of commentary but the reader is unable to validate the claims as no specific data is provided to demonstrate the efficacy/selectivity etc of the flavonoid compounds that are discussed.

Response 4:

Thank you very much for your advice. We conducted more articles and added data content especially for quercetin on page 4 line 197-198 and page 6 line 328-330.

What`s more,some studies show that the IC50 value of quercetin against MCF-7 cells was 0.87mg/mL[54]

In addition, it has been reported that when IC50 of quercetin is 4.88mg/mL, the survival rate of HepG2 cell lines can be best reduced[54].

[54]Ahmed,H.; Moawad,A.;Owis,A.;AbouZid,S.;Ahmed,O. Flavonoids of Calligonum polygonoides and their cytotoxicity. Pharm Biol 2016 ,54,2119-2126.

Point 5:Table 1 contains some interesting information but the section around it is not well developed.

Response 5:

Thank you so much.We added the classification of on flavonoids in the Table 1to make it clearer, and we also added the presumed possible mechanism based on the “pathway” and “Main Effect”.And we found that “Flavonoids regulate the expression of vascular growth factor and block its signal transduction. In addition to these mechanisms, flavonoids could regulate cell apoptosis through the activation of mitogen activated protein kinase (MAPK) signal transduction pathway and protein kinase C(PKC), which also could inhibit the expression of COX-2 and reduce the induced PGE2.[104] ”one page 7 line 287-292.

Flavonoid

Class 

Model

Pathways

Main Effect

References

lung cancer

Peonidin 3-Glucoside

Anthocyanidine

cell

LOX

inducing apoptosis

[21]

Kaempferol

Flavanonole

cell

MMP-2

inducing apoptosis

[23]

Epimedium flavin

cell

PTEN/AKT

inducing apoptosis

[24]

Quercetin

cell

NF-κB/MMP

inducing apoptosis

[25]

Luteolin

Flavone

cell

ERK /MAPK

MAPK pathways activation

[26] [27] [28] [29]

Luteolin

cell

NF-kB (p65)

inducing TNF-mediated apoptotic cell death

[30]

Apigenin

cell

ROS

inducing apoptosis

[34]

Naringenin

Flavanone

cell

AKT/MMP

inhibiting tumor growth and metastasis

[35][36]

Epigallocatechin-3-gallate

Flavanole

cell

AP-1, MAPK,NF-κB, and COX-2

inhibiting tumor growth and metastasis

[38]

Baicalein

Flavone

cell

12-LOX

inducing apoptosis

[22]

Chrysin

cell

ROS

inducing oxidative stress

[31]

Breast cancer

Ampelopsin

Flavanonols

cell

ROS

promoting ER stress

[40]

Wogonin

cell

ERK

inducing apoptosis

[41]

Isoliquiritigenin

Chalkone

cell

COX-2/PI3K, PDK, Akt

inhibiting tumor growth and metastasis

[42]

Naringenin

Flavanone

cell

ERK

inhibiting tumor growth

[50]

Hesperetin

cell

ROS

inducing apoptosis

[44]

Daidzein

Isoflavone

cell

MMP

inhibiting tumor growth and metastasis

[45]

EGCG

Flavanole

cell

MMP

inhibiting tumor growth and metastasis

[46]

Kaempferol

Flavanonole

cell

PLK-1/ERK

inhibiting tumor growth and inducing apoptosis

[47][48][49]

Synthesized flavonoid LW-214

Flavone

cell

ROS

Induce cell apoptosis through mitochondrial pathway

[51]

Synthesized flavonoid Oncamex

Unkown

cell

ROS

Induce cell apoptosis through mitochondrial pathway

[52]

Synthesized  flavonoid LGF-500

Flavone

cell

ROS/RNS

inducing apoptosis

[53]

gastric cancer

Naringenin

Flavanone

cell

MMP

inhibiting chemical-induced cell invasion,metastasis

[58]

Hesperetin

cell

ROS

inhibiting cell proliferation and inducing apoptosis

[55]

Genistein

Isoflavone

cell

NF-κB/COX-2

inhibiting angiogenesis and metastasis

[56]

Genistein

mouse

NF-κB

suppressing mortality, tumor number, tumor burden and chemical-induced inflammatory responses

[57]

Colorectal

EGCG

Flavanole

cell

ERK1/2/p38MAPK

inducing apoptosis

[59]

Quercetin

Flavanonole

cell

NF-κB

inducing apoptosis

[62]

Quercetin

cell

AMPK/p53

inducing apoptosis

[63]

Baicalein

Flavone

cell

MMP-2/MMP-9/AKT

inhibiting cell migration and invasion

[64]

Genistein

Isoflavone

cell

NF-κB

inducing apoptosis

[65]

Naringenin

Flavanone

cell

NF-κB/ p65

inducing apoptosis and cell cycle arrest

[66]

Liver Cancer

Oroxin B

Flavone

cell

PTEN/PI3K/AKT Pathway

fighting against liver cancer

[67]

Baicalein

cell

PI3K/Akt pathway

 inhibiting the proliferation of Bel-7402 cells

[69]

Baicalein

cell

MAPK and NF-kB pathway 、ERK pathway 、AKT/mTOR pathway

fighting against hepatocellular carcinoma

[70]

Isobavachalcone

Flavanonole

cell

 ERKs/RSK2 signaling pathway

anti‐proliferative and pro‐apoptotic effects on human liver cancer cells

[71]

Fisetin

cell

 TGF-β1/ERK signaling pathway

down-regulated in liver cancer cells

[72]

Xanthohumol

Chalkone

cell

NF-κB/p53

inducing apoptosis, modulating the NF-κB/p53 and the Notch1 signaling pathways

[75]

Quercetin

Flavanonole

cell

ERK 

inducing apoptosis

[76]

Quercetin

cell

suppressing chemical-induced carcinogenesis

[77]

Cervical Cancer

Apigenin

Flavone

cell

ROS and LPO

oxidative stress

[78]

Quercetin

Flavanonole

cell

MAPK

decreasing cell proliferation,invasion, angiogenesis

[79]

Genistein

Isoflavone

cell

MMP-9

inducing apoptosis, cell cycle arrest,suppressing cell migration

[80]

Xanthohumol

Chalkone

cell

NF-κB

decreasing expression of CXCR4, inhibiting cell invasion induced by CXCL12

[81][82]

Prostate

Xanthohumol

Chalkone

mouse

FAK/AKT/ROS

suppressing tumor growth and progression

[84]

Apigenin

Flavone

mouse

PI3K/Akt/FoxO

suppressing tumor growth,angiogenesis, metastasis

[85]

Quercetin

Flavanonole

mouse

peroxides and H2O2 got decreased

inhibiting carcinogenesis induced byhormone and carcinogen

[86]

Quercetin

mouse

PI3K/Akt or RAF/MEK/ERK

inhibiting carcinogenesis induced byhormone and carcinogen

[87]

Apigenin

Flavone

cell

NF-κB/AKT

inducing apoptosis, inhibiting cell invasion, motility

[88]

Naringenin

Flavanone

cell

ERK

invasion and migration

[89]

Ovarian cancer

Apigenin

Flavone

cell

MMP

Carcinogenesis

[90]

Esophageal cancer

Hesperetin

Flavanone

cell/mouse

MMP

inhibiting chemical-induced cell invasion,metastasis,

[91]

Osteosarcoma

Phellamurin

Flavanonols

cell

MMP

inhibiting tumor growth and metastasis

[92]

Point 6:There are no figures whatsoever and this is a significant weakness of the paper.

Response 6:

Thank your for your good advice, we had added a figure of mechanism on page 8 as followed.

Point 7:What are the conclusions? And future prospects? These sections need to be included, and they need to be robust

Response 7:

 Thank you very much for your suggestions. The conclusions on page 7 Line 284-292 are “ Tumors need a rich blood supply to grow and survive. The newborn blood vessels can make tumor cells along the basal membrane defect space and the substrate to grow around, and MMPs expression changes, especially the MMP-2 and MMP-9 higher expression, basement membrane degradation of collagen type Ⅳ ability enhancement is the key step in the cancer cell invasion and metastasis, a lot of flavonoids of MMP-2 and (or) MMP-9 has inhibitory effect.Flavonoids regulate the expression of vascular growth factor and block its signal transduction. In addition to these mechanisms, flavonoids could regulate cell apoptosis through the activation of mitogen activated protein kinase (MAPK) signal transduction pathway and protein kinase C(PKC), which also could inhibit the expression of COX-2 and reduce the induced PGE2.[104]” We also have added discussion on page8 to show our views on page 8 Line 295-306 as followed.

4.Discussion

It can be seen that flavonoids can regulate the inflammatory response and oxidative stress of tumors to achieve anticancer effect. However, there are few clinical cases of using flavonoids to treat cancer. Firstly, the oral absorption of flavonoids is slow, the bioavailability is low, and the existing preparation technology of ginkgo flavonoid pellets is in the research and development stage. Secondly, the flavonoid extract is unstable, and it needs to be configured with other complexes. For example, myricetin exhibited substantial limitations, such as poor water-solubility, and low stability in body when it was administrated by oral.M10 was produced by adding a hydrophilic glycosylation group and then forming a sodium salt derivative, which exhibited excellent water-solubility (>100 mg/mL), and better stability in Wistar rat plasma and liver microsomes.[ 10 5]类黄酮药物将通过技术创新在临床中使用,这可以减轻西药的毒性和副作用以及耐药性。

非常感谢您给我们这次难得的机会,并指出了我们手稿中的这些错误以使我们学到更多。

真诚的

Reviewer 2 Report

This review article summarizes effects of flavonoids various cancer cell lines concisely. Although the structure – activity relationship for this article is very important, it lacks structural formulae. They should be given.

Additionally, the authors should consider the following points.

1) Names of types of flavonoids, i.e., flavonols, flavanones, isoflavones (as shown in lines 63-67) should be added for respective compounds.

2) Some compounds such as gallic acid, ellagic acid, and resveratrol are not flavonoids but polyphenolic compounds.

3) All of the compound names should be given in lower case characters except for tops of sentences: For example, “Genistein” in line 124 should be “genistein.”; “Quercetin” in line 230 should be “quercetin.”

4) Although the compound name “PFEs” is given in Table 1 for reference 83, this paper treats gallic acid.

Author Response

Dear Sir or Madam,

Thank you so much for carefully reviewing and reconsidering our manuscript. And we would like to express our sincere thanks to the reviewers for the constructive and positive comments.

Answers to reviewers:

Reviewer #2:

Point 1:Names of types of flavonoids, i.e., flavonols, flavanones, isoflavones (as shown in lines 63-67) should be added for respective compounds.

Response 1:

Thank you sir or ma`am, we reclassified the flavonoids on the table 1 in red and corrected some spelling mistakes on page 8-12 line 307-308 which are as followed.

Flavonoid

Class 

Model

Pathways

Main Effect

References

lung cancer

Peonidin 3-Glucoside

Anthocyanidine

cell

LOX

inducing apoptosis

[21]

Kaempferol

Flavanonole

cell

MMP-2

inducing apoptosis

[23]

Epimedium flavin

cell

PTEN/AKT

inducing apoptosis

[24]

Quercetin

cell

NF-κB/MMP

inducing apoptosis

[25]

Luteolin

Flavone

cell

ERK /MAPK

MAPK pathways activation

[26] [27] [28] [29]

Luteolin

cell

NF-kB (p65)

inducing TNF-mediated apoptotic cell death

[30]

Apigenin

cell

ROS

inducing apoptosis

[34]

Naringenin

Flavanone

cell

AKT/MMP

inhibiting tumor growth and metastasis

[35][36]

Epigallocatechin-3-gallate

Flavanole

cell

AP-1, MAPK,NF-κB, and COX-2

inhibiting tumor growth and metastasis

[38]

Baicalein

Flavone

cell

12-LOX

inducing apoptosis

[22]

Chrysin

cell

ROS

inducing oxidative stress

[31]

Breast cancer

Ampelopsin

Flavanonols

cell

ROS

promoting ER stress

[40]

Wogonin

cell

ERK

inducing apoptosis

[41]

Isoliquiritigenin

Chalkone

cell

COX-2/PI3K, PDK, Akt

inhibiting tumor growth and metastasis

[42]

Naringenin

Flavanone

cell

ERK

inhibiting tumor growth

[50]

Hesperetin

cell

ROS

inducing apoptosis

[44]

Daidzein

Isoflavone

cell

MMP

inhibiting tumor growth and metastasis

[45]

EGCG

Flavanole

cell

MMP

inhibiting tumor growth and metastasis

[46]

Kaempferol

Flavanonole

cell

PLK-1/ERK

inhibiting tumor growth and inducing apoptosis

[47][48][49]

Synthesized flavonoid LW-214

Flavone

cell

ROS

Induce cell apoptosis through mitochondrial pathway

[51]

Synthesized flavonoid Oncamex

Unkown

cell

ROS

Induce cell apoptosis through mitochondrial pathway

[52]

Synthesized  flavonoid LGF-500

Flavone

cell

ROS/RNS

inducing apoptosis

[53]

gastric cancer

Naringenin

Flavanone

cell

MMP

inhibiting chemical-induced cell invasion,metastasis

[58]

Hesperetin

cell

ROS

inhibiting cell proliferation and inducing apoptosis

[55]

Genistein

Isoflavone

cell

NF-κB/COX-2

inhibiting angiogenesis and metastasis

[56]

Genistein

mouse

NF-κB

suppressing mortality, tumor number, tumor burden and chemical-induced inflammatory responses

[57]

Colorectal

EGCG

Flavanole

cell

ERK1/2/p38MAPK

inducing apoptosis

[59]

Quercetin

Flavanonole

cell

NF-κB

inducing apoptosis

[62]

Quercetin

cell

AMPK/p53

inducing apoptosis

[63]

Baicalein

Flavone

cell

MMP-2/MMP-9/AKT

inhibiting cell migration and invasion

[64]

Genistein

Isoflavone

cell

NF-κB

inducing apoptosis

[65]

Naringenin

Flavanone

cell

NF-κB/ p65

inducing apoptosis and cell cycle arrest

[66]

Liver Cancer

Oroxin B

Flavone

cell

PTEN/PI3K/AKT Pathway

fighting against liver cancer

[67]

Baicalein

cell

PI3K/Akt pathway

 inhibiting the proliferation of Bel-7402 cells

[69]

Baicalein

cell

MAPK and NF-kB pathway 、ERK pathway 、AKT/mTOR pathway

fighting against hepatocellular carcinoma

[70]

Isobavachalcone

Flavanonole

cell

 ERKs/RSK2 signaling pathway

anti‐proliferative and pro‐apoptotic effects on human liver cancer cells

[71]

Fisetin

cell

 TGF-β1/ERK signaling pathway

down-regulated in liver cancer cells

[72]

Xanthohumol

Chalkone

cell

NF-κB/p53

inducing apoptosis, modulating the NF-κB/p53 and the Notch1 signaling pathways

[75]

Quercetin

Flavanonole

cell

ERK 

inducing apoptosis

[76]

Quercetin

cell

suppressing chemical-induced carcinogenesis

[77]

Cervical Cancer

Apigenin

Flavone

cell

ROS and LPO

oxidative stress

[78]

Quercetin

Flavanonole

cell

MAPK

decreasing cell proliferation,invasion, angiogenesis

[79]

Genistein

Isoflavone

cell

MMP-9

inducing apoptosis, cell cycle arrest,suppressing cell migration

[80]

Xanthohumol

Chalkone

cell

NF-κB

decreasing expression of CXCR4, inhibiting cell invasion induced by CXCL12

[81][82]

Prostate

Xanthohumol

Chalkone

mouse

FAK/AKT/ROS

suppressing tumor growth and progression

[84]

Apigenin

Flavone

mouse

PI3K/Akt/FoxO

suppressing tumor growth,angiogenesis, metastasis

[85]

Quercetin

Flavanonole

mouse

peroxides and H2O2 got decreased

inhibiting carcinogenesis induced byhormone and carcinogen

[86]

Quercetin

mouse

PI3K/Akt or RAF/MEK/ERK

inhibiting carcinogenesis induced byhormone and carcinogen

[87]

Apigenin

Flavone

cell

NF-κB/AKT

inducing apoptosis, inhibiting cell invasion, motility

[88]

Naringenin

Flavanone

cell

ERK

invasion and migration

[89]

Ovarian cancer

Apigenin

Flavone

cell

MMP

Carcinogenesis

[90]

Esophageal cancer

Hesperetin

Flavanone

cell/mouse

MMP

inhibiting chemical-induced cell invasion,metastasis,

[91]

Osteosarcoma

Phellamurin

Flavanonols

cell

MMP

inhibiting tumor growth and metastasis

[92]

Point 2:Some compounds such as gallic acid, ellagic acid, and resveratrol are not flavonoids but polyphenolic compounds.

Response 2:

Thank you for pointing out our mistakes. We had deleted the polyphenolic compounds which mentioned in the article and also check the similar mistakes in the manuscript, and modified them, thanks again.

Point 3:All of the compound names should be given in lower case characters except for tops of sentences: For example, “Genistein” in line 124 should be “genistein.”; “Quercetin” in line 230 should be “quercetin.”

Response 3:

 Thank you very much for your careful reviewing, and We have changed the characters already, moreover, we also check similar corrections in the whole manuscript and modified them in red.

Point 4:Although the compound name “PFEs” is given in Table 1 for reference 83, this paper treats gallic acid.

Response 4:

Thank you for your careful reviewing. We had deleted the PFEs and gallic acid which is not belongs to flavonoids, thanks again.

Thank you very much for giving us this precious chance and pointing out these mistakes in our manuscript to let us learning more.

Sincerely

Round 2

Reviewer 1 Report

The authors have addressed my queries to a good level and have added additional points to the manuscript. It is now acceptable for publication. However, the order of the manuscript is now confusing, eg they have placed a discussion section and Table after the conclusions and it is unclear why this has been done. I therefore feel the manuscript needs some in house editing, to maximise the logical flow of the paper, before it can be published

Reviewer 2 Report

The manuscript was revised adequately, and now this reviewer recommends publishing it in the journal.